# True S-cones are concentrated in the ventral mouse retina and wired for color detection in the upper visual field

Francisco M Nadal-Nicolás[1]*, Vincent P Kunze[1†], John M Ball[1†], Brian T Peng[1†], Akshay Krishnan[1†], Gaohui Zhou[1†], Lijin Dong[2], Wei Li[1]*

[1]Retinal Neurophysiology Section, National Eye Institute, National Institutes of Health, Bethesda, United States; [2]Genetic Engineering Facility, National Eye Institute, National Institutes of Health, Bethesda, United States

**Abstract** Color, an important visual cue for survival, is encoded by comparing signals from photoreceptors with different spectral sensitivities. The mouse retina expresses a short wavelength-sensitive and a middle/long wavelength-sensitive opsin (S- and M-opsin), forming opposing, overlapping gradients along the dorsal-ventral axis. Here, we analyzed the distribution of all cone types across the entire retina for two commonly used mouse strains. We found, unexpectedly, that 'true S-cones' (S-opsin only) are highly concentrated (up to 30% of cones) in ventral retina. Moreover, S-cone bipolar cells (SCBCs) are also skewed towards ventral retina, with wiring patterns matching the distribution of true S-cones. In addition, true S-cones in the ventral retina form clusters, which may augment synaptic input to SCBCs. Such a unique true S-cone and SCBC connecting pattern forms a basis for mouse color vision, likely reflecting evolutionary adaptation to enhance color coding for the upper visual field suitable for mice's habitat and behavior.

*For correspondence:
nadalnicolasfm@nih.gov (FMN-N);
liwei2@nei.nih.gov (WL)

†These authors contributed equally to this work

Competing interests: The authors declare that no competing interests exist.

## Introduction

Topographic representation of the visual world in the brain originates from the light-sensitive photo-receptors in the retina (*Rhim et al., 2017*). Although the neuronal architecture of the retina is similar among different vertebrates, the numbers and distributions of photoreceptors vary considerably (*Hunt and Peichl, 2014*). Such patterns have been evolutionarily selected, adapting to the animal's unique behavior (diurnal or nocturnal) and lifestyle (prey or predator) for better use of the visual information in the natural environment (*Dominy and Lucas, 2001*; *Gerl and Morris, 2008*; *Peichl, 2005*). Color, an important visual cue for survival, is encoded by comparing signals carried by photoreceptors with different spectral preferences (*Baden and Osorio, 2019*). While amongst mammals, trichromatic color vision is privileged for some primates (*Jacobs et al., 1996*; *Nathans et al., 1986*; *Yokoyama and Yokoyama, 1989*), most terrestrial mammals are dichromatic (*Marshak and Mills, 2014*; *Puller and Haverkamp, 2011*; *Jacobs, 1993*). The mouse retina expresses two types of cone opsins, S- and M-opsin, with peak sensitivities at 360 nm and 508 nm, respectively (*Jacobs et al., 1991*; *Nikonov et al., 2006*). The expression patterns of these two opsins form opposing and overlapping gradients along the dorsal-ventral axis, resulting in a majority of cones expressing both opsins (herein either 'mixed cones' or M+S+) (*Applebury et al., 2000*; *Ng et al., 2001*; *Wang et al., 2011*). Thus, S-opsin enrichment in the ventral retina better detects short-wave-length light from the sky, and M-opsin in the dorsal retina perceives the ground (e.g., a grassy field) (*Baden et al., 2013*; *Gouras and Ekesten, 2004*; *Osorio and Vorobyev, 2005*; *Szél et al., 1992*), while co-expression of both opsins (herein either mixed cones or M+S+) (*Röhlich et al., 1994*)

**eLife digest** Many primates, including humans, can see color better than most other mammals. This difference is due to the variety of light-detecting proteins – called opsins – that are produced in the eye by cells known as cones. While humans have three, mice only have two different opsins, known as S and M, which detect blue/UV and green light, respectively. Mouse cones produce either S-opsins, M-opsins or both. Fewer than 10 percent of cone cells in mice produce just the S-opsin, and these cells are essential for color vision.

Mice are commonly used in scientific research, and so their vision has been well studied. However, previous research has produced conflicting results. Some studies report that cone cells that contain only S-opsin are evenly spread out across the retina. Other evidence suggests that color vision in mice exists only for the upper field of their vision, in other words, that mice can only distinguish colors that appeared above them.

Nadal-Nicolás et al. set out to understand how to reconcile these contrasting findings. Molecular tools were used to detect S- and M-opsin in the retina of mice and revealed large differences between the lower part, known as the ventral retina, and the upper part, known as the dorsal retina. The ventral retina detects light coming from above the animal, and about a third of cone cells in this region produced exclusively S-opsin, compared to only 1 percent of cones in the dorsal retina. These S-opsin cone cells in the ventral retina group into clusters, where they connect with a special type of nerve cells that transmit this signal.

To better understand these findings, Nadal-Nicolás et al. also studied albino mice. Although albino mice have a different distribution of S-opsin protein in the retina, the cone cells producing only S-opsin are similarly clustered in the ventral retina. This suggests that the concentration of S-opsin cone cells in the ventral retina is an important feature in mouse sight. This new finding corrects the misconception that S-opsin-only cone cells are evenly spread throughout the retina and supports the previous evidence that mouse color vision is greatest in the upper part of their field of vision. Nadal-Nicolás et al. suggest this arrangement could help the mice to detect predators that may attack them from above during the daytime. Together, these new findings could help to improve the design of future studies involving vision in mice and potentially other similar species.

broadens the spectral range of individual cones and improves perception under varying conditions of ambient light (*Chang et al., 2013*).

This unusual opsin expression pattern poses a challenge for color-coding, particularly so for mixed cones. However, it has been discovered that a small population of cones only expresses S-opsin ('true S-cones', or $S^+M^-$). These true S-cones are thought to be evenly distributed across the retina (*Franke et al., 2019*; *Haverkamp et al., 2005*; *Szatko et al., 2019*; *Wang et al., 2011*) and to be critical for encoding color, especially in the dorsal retina where they are quasi-evenly distributed in a sea of cones expressing only M-opsin ($M^+S^-$), a pattern akin to mammalian retinas in general (*Haverkamp et al., 2005*; *Wang et al., 2011*). Nonetheless, subsequent physiological studies revealed that color-opponent retinal ganglion cells (RGCs) are more abundant in the dorsal-ventral transition zone (*Chang et al., 2013*) and the ventral retina (*Joesch and Meister, 2016*). Recent large scale two-photon imaging results further demonstrated that color opponent cells were mostly located in the ventral retina (*Szatko et al., 2019*). Intriguingly, a behavior-based mouse study demonstrated that their ability to distinguish color is also restricted to the ventral retina (*Denman et al., 2018*). These results prompt us to study, at the single-cell level and across the whole retina, the spatial distributions of cone types with different opsin expression configurations and, more importantly, with regard to S-cone bipolar cell connections in order to better understand the anatomical base for the unique color-coding scheme of the mouse retina.

## Results and discussion

### True S-cones are highly concentrated in the ventral retina of pigmented mouse

In mouse retina, the gradients of S- and M-opsin expression along the dorsal-ventral axis have been well documented (*Figure 1A–B*; *Applebury et al., 2000*; *Calderone and Jacobs, 1995*; *Chang et al., 2013*; *Haverkamp et al., 2005*; *Jelcick et al., 2011*; *Lyubarsky et al., 1999*; *Ortín-Martínez et al., 2014*; *Szél et al., 1992*; *Wang et al., 2011*), but the distribution of individual cone types with different combinations of opsin expression across the whole retina has not been characterized (but see *Baden et al., 2013*; *Eldred et al., 2020*), which we discuss below. We developed a highly reliable algorithm to automatically quantify the different opsins (S and M) and cone types

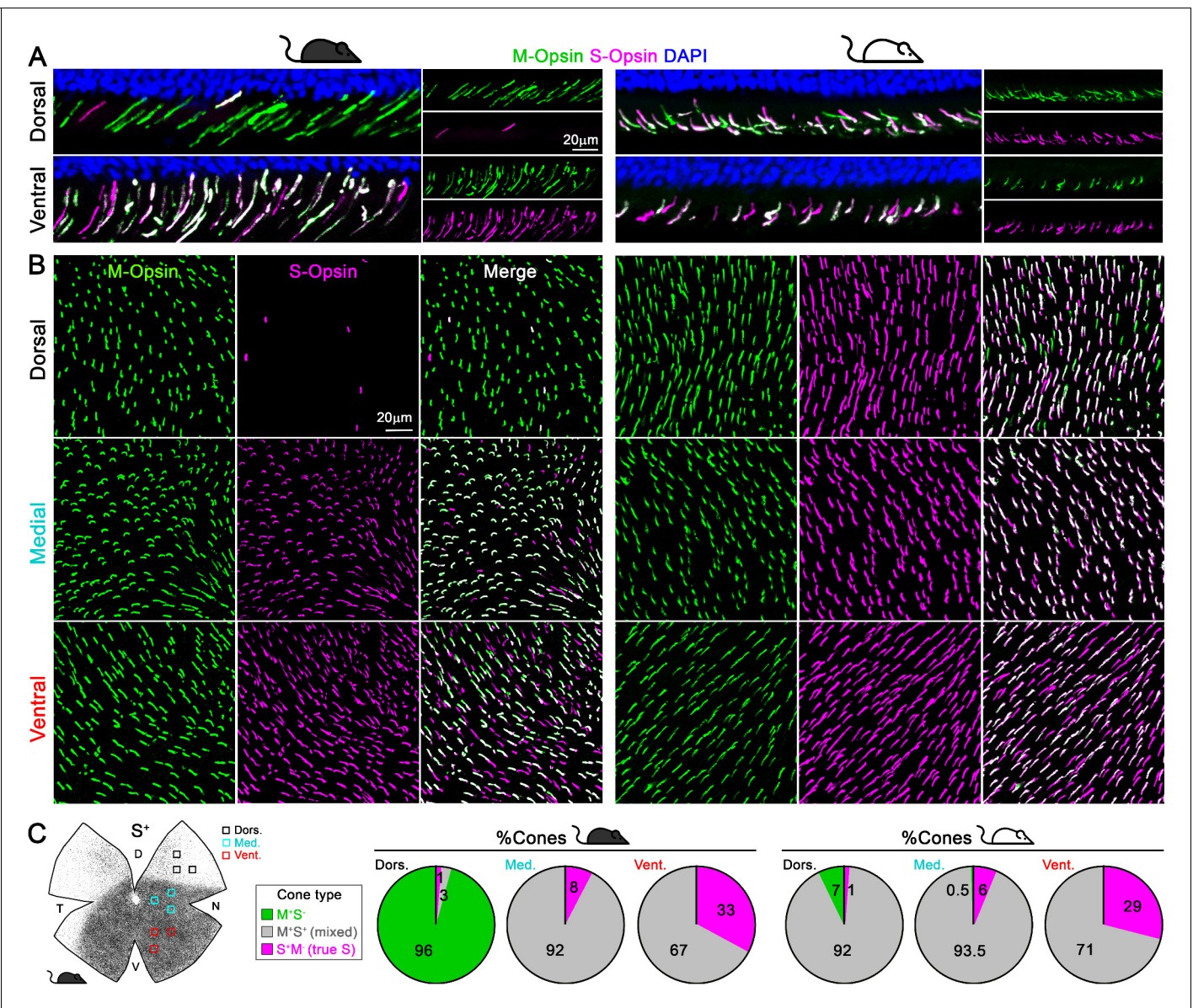

**Figure 1.** Cone outer segments across retinal areas. Immunodetection of M and S wavelength-sensitive opsins in retinal sections (**A**) and flat-mount retinas (**B**) in two mouse strains (pigmented and albino mice, left and right columns respectively). (**C**) Retinal scheme of S-opsin expression used for image sampling to quantify and classify cones in three different retinal regions. Pie graphs showing the percentage of cones manually classified as $M^+S^-$ (green), $S^+M^-$ (true S, magenta) and $M^+S^+$ (mixed, gray) based on the opsin expression in different retinal areas from four retinas per strain. Black mouse: pigmented mouse strain (C57BL6), white mouse: albino mouse strain (CD1).

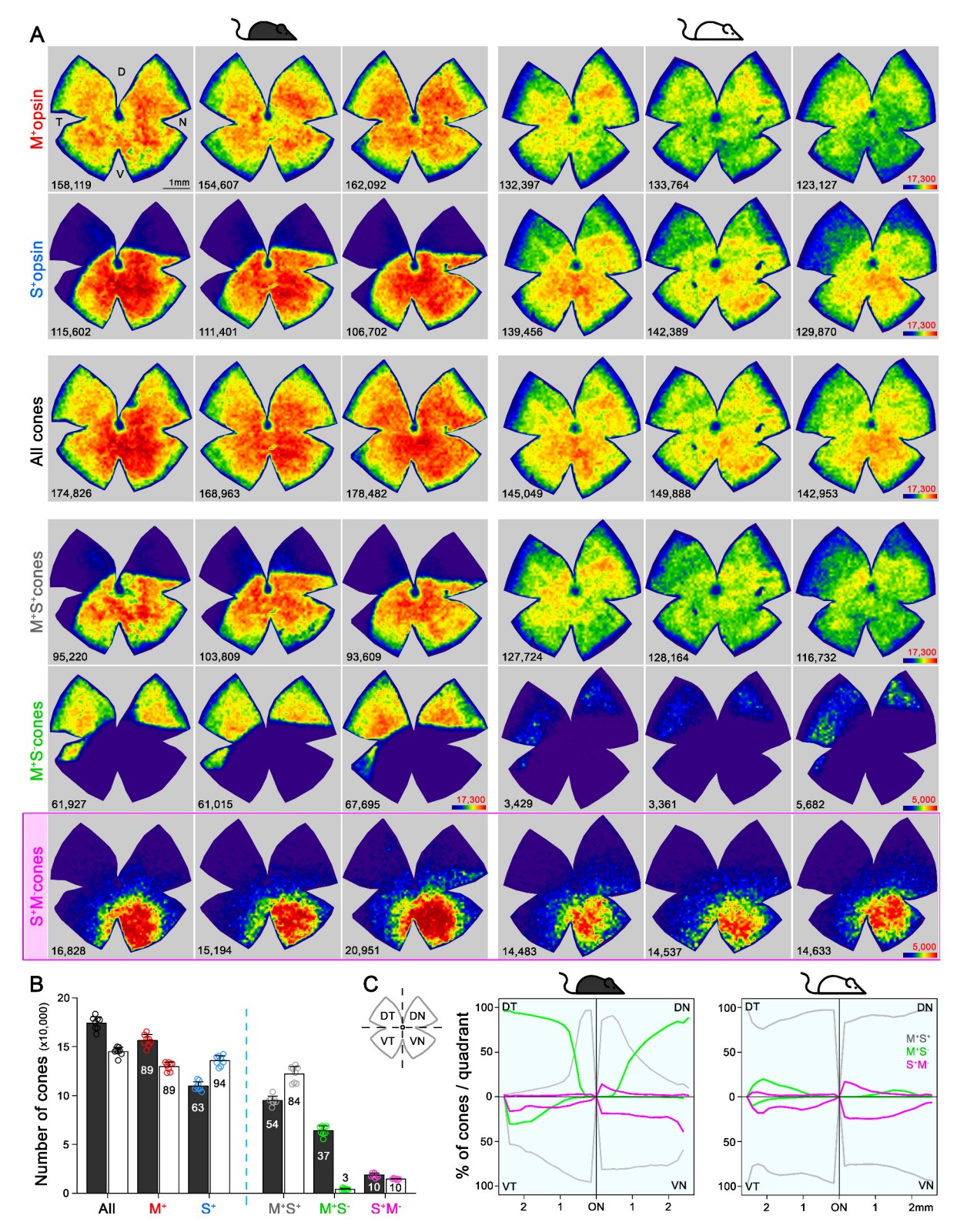

**Figure 2.** Topography and total number of different opsins ($M^+$, $S^+$) and cone-type populations in the whole mouse retina. (**A**) Density maps depicting the distributions of different opsins expressing cones ($M^+$ and $S^+$) and different cone populations classified anatomically as: All, $M^+S^+$ (mixed), $M^+S^-$, $S^+M^-$ (true S) cones in pigmented and albino mice (left and right side respectively). Each column shows different cone populations from the same retina and, at the bottom of each map is shown the number of quantified cones. Color scales are shown in the right panel of each row (from 0 [purple] to

*Figure 2 continued on next page*

Figure 2 continued
17,300 [dark red] for all cone types except to 5000 cones/mm² [dark red] for the true S-cones and M⁺S⁻-cone in the albino strain). Retinal orientation depicted by D: dorsal, N: nasal, T: temporal, V: ventral. (B) Histogram showing the mean ± standard deviation of different cone subtypes for eight retinas per strain (*Supplementary file 1B*). The percentages of each cone subtype are indicated inside of each bar, where 100% indicates the total of the 'all cones' group. (C) Opsin expression profile across the different retinal quadrants (retinal scheme, DT: dorsotemporal, DN: dorsonasal, VT: ventrotemporal, VN: ventronasal). Line graphs show the spatial profile of relative opsins expression (mixed [gray], M⁺S⁻ [green], true S-cones [magenta]), where the sum of these three cone populations at a given distance from the optic nerve (ON) head equals 100%. Black mouse: pigmented mouse strain, white mouse: albino mouse strain.

The online version of this article includes the following figure supplement(s) for figure 2:

**Figure supplement 1.** Validation of automatic routine for cone outer segment quantification.

(M⁺S⁻, true S, and mixed cones, *Figure 2*, *Figure 2—figure supplement 1*) based on high-resolution images of entire flat-mount retinas immunolabeled with S- and M-opsin antibodies (*Figure 2—figure supplement 1*). As demonstrated in examples of opsin labeling from dorsal, medial, and ventral retinal areas of the pigmented mouse (*Figure 1B*, left), while M opsin-expressing cones (M⁺: M⁺S⁺ + M⁺S⁻) were relatively evenly distributed across three regions, S opsin-expressing cones (S⁺: M⁺S⁺ + S⁺M⁻) showed considerable anisotropy, with a high density in the ventral retina and a precipitous drop in the dorsal retina, confirming previous observations (*Haverkamp et al., 2005*; *Jelcick et al., 2011*; *Ortín-Martínez et al., 2014*). Surprisingly, instead of finding an even distribution of true S-cones as previously presumed (*Baden et al., 2013*; *Haverkamp et al., 2005*; *Wang et al., 2011*), we found the ventral region had much more numerous true S-cones (~30% of the local cone population; *Figure 1C* left, *Supplementary file 1A*) than did the dorsal region (~1%). This result is evident from density maps of cone types from three examples of pigmented mice, showing highly concentrated true S-cones in the ventral retina (*Figure 2A*, left column, bottom row). In addition, M⁺S⁻-cones were concentrated in the dorsal retina, whereas mixed cones dominated the medial and ventral retina (*Figure 1C* left and *Figure 2A*, left column, 4th and 5th rows).

## Despite the vast difference in S-opsin expression pattern, the distribution of true S-cones is strikingly similar between the pigmented and albino mouse

Such a highly skewed distribution of true S-cones conflicts with the general notion that true S-cones only account for ~5% of cones and are evenly distributed across the mouse retina (*Baden et al., 2013*; *Franke et al., 2019*; *Haverkamp et al., 2005*; *Szatko et al., 2019*; *Wang et al., 2011*); however, it is not unprecedented considering the diverse S-cone patterns seen in mammals (*Ahnelt et al., 2000*; *Ahnelt and Kolb, 2000*; *Calderone et al., 2003*; *Hendrickson et al., 2000*; *Hendrickson and Hicks, 2002*; *Kryger et al., 1998*; *Müller and Peichl, 1989*; *Nadal-Nicolás et al., 2018*; *Ortín-Martínez et al., 2014*; *Ortín-Martínez et al., 2010*; *Peichl, 2005*; *Schiviz et al., 2008*; *Szél et al., 2000*). Therefore, we also examined an albino mouse line to determine whether this observation persists across different mouse strains. Overall, albino retinas had slightly smaller cone populations (*Figure 2B*, *Supplementary file 1B*; *Ortín-Martínez et al., 2014*). Interestingly, while M-opsin expressing cones had similar distributions in both strains, S-opsin expression extended well into the dorsal retina of the albino mouse, exhibiting a greatly reduced gradient of S-opsin expression toward the dorsal retina compared to that seen in pigmented mice (*Figure 1B–C*, *Figure 2A* second row; *Applebury et al., 2000*; *Ortín-Martínez et al., 2014*). Consequently, most cones in the dorsal retina were mixed cones, and M⁺S⁻ cones were very sparse (7%, compared to 97% in pigmented mouse, *Figure 1C* right, *Supplementary file 1A*, *Figure 2A* right). However, despite these differences, the percentage and distribution of true S-cones were remarkably conserved between strains. In both strains, true S-cones were extremely sparse in the dorsal retina (1%) but highly concentrated in the ventral retina (33% vs 29%, *Figure 1C* and *Supplementary file 1A*). Notably, the density maps of true S-cones are nearly identical in both strains (*Figure 2A*, bottom row). Evaluating the distribution of three main cone populations (mixed, M⁺S⁻, and true S-cone) in four retinal quadrants centered upon the optic nerve head reveals different profiles between pigmented and albino strain for mixed and M⁺S⁻ cones (*Figure 2C*). For example, in the dorsotemporal (DT) quadrant, we observed an increase of M⁺S⁻ cones from the center to the periphery (green line) in pigmented mice, compared to a majority of mixed cones (gray line) in albino mice. However, true S-cone

profiles (magenta lines) were similar between the two strains in all quadrants, except for a slightly increased density along the edge of the ventronasal (VN) quadrant in pigmented mice. A recent study successfully modeled cone opsin expression and type determination according to graded thyroid hormone signaling in a pigmented mouse strain (C57BL/6) (*Eldred et al., 2020*). It would be interesting to see whether a different pattern of thyroid hormone and/or receptor distribution could recapitulate a similar true S-cone distribution with a very different form of S-opsin expression.

## S-cone bipolar cells exhibit a dorsal-ventral gradient with a higher density in the ventral retina

One major concern regarding cone classification based on opsin immunolabeling is that some $S^+M^-$ cones may instead be mixed cones with low M-opsin expression (*Applebury et al., 2000*; *Baden et al., 2013*; *Nikonov et al., 2006*; *Röhlich et al., 1994*). Even though a similar cone-type distributions have been observed in mouse retina, it has been assumed that only a fraction of the $S^+M^-$ cones are 'true' S-cones (*Baden et al., 2013*; *Eldred et al., 2020*). Out of caution, $S^+M^-$ cones were only referred to as 'anatomical' S-cones due to a lack of confirmation regarding their bipolar connections (*Baden et al., 2013*). Thus, both true S-cones and S-cone bipolar cells have been generally acknowledged to be evenly distributed across the retina (*Haverkamp et al., 2005*; *Wang et al., 2011*; *Baden et al., 2013*; *Szatko et al., 2019*; *Franke et al., 2019*; *Eldred et al., 2020*). In order to confirm the distribution of true S-cones, it is critical to uncover the distribution and dendritic contacts of S-cone bipolar cells (type 9, or SCBCs). Previously, SCBCs have only been identified among other bipolar, amacrine and ganglion cells in a Thy1-Clomeleon mouse line, rendering the quantification of their distribution across the entire retina impractical (*Haverkamp et al., 2005*). We generated a Copine9-Venus mouse line, in which SCBCs are specifically marked (*Figure 2—figure supplement 1C*), owing to the fact that *Cpne9* is an SCBC-enriched gene (*Shekhar et al., 2016*). In retinal sections, these Venus$^+$ bipolar cells have axon terminals narrowly ramified in sub-lamina 5 of IPL (*Figure 3A*), closely resembling type 9 BCs as identified in EM reconstructions (*Behrens et al., 2016*; *Stabio et al., 2018a*). In flat-mount view, these bipolar cells are often seen to extend long dendrites to reach true S-cones, bypassing other cone types (*Figure 3B–C*). The majority of dendritic endings formed enlarged terminals beneath true S-cones pedicles (*Figure 3C–c'*), but occasional slender 'blind' endings were present (arrow in *Figure 3C–c''*), which have been documented for S-cone bipolar cells in many species (*Haverkamp et al., 2005*; *Herr et al., 2003*; *Kouyama and Marshak, 1992*). Unexpectedly, we found that the distribution of SCBCs was also skewed toward VN retina, albeit with a shallower gradient (*Figure 3D–E*). To examine the connections between true S-cones and SCBCs, we immunolabeled S- and M-opsins in Copine9-Venus mouse retinas. Because M-opsin antibody signals did not label cone structures other than their outer segments, we first identified true S-cones at the outer segment level and then traced S-opsin labeling to their pedicles in the outer plexiform layer (OPL), where they connect with SCBCs (*Figure 3C*, for more details see material and methods). Although convergent as well divergent connections were found between true S-cones and SCBCs in both dorsal and ventral retina (see the source data), we noted different connectivity patterns. While in the dorsal retina, a single true S-cone connected to approximately 4 SCBCs (3.8 ± 0.2, see material and methods), in the ventral retina, a single SCBC contacted approximately 5 true S-cones (4.6 ± 0.4; *Figure 3C*, *Supplementary file 2*). These results agree well with the true S-cone to SCBC ratios calculated from cell densities in the DT and VN retina. Specifically, in the dorsal retina, the true S-cone to SCBC ratio was approximately 1:3.6, compared to 5.3:1 in the ventral retina (*Supplementary file 3*). Accordingly, both data sets support the presence of a prevalent divergence of true S-cone to SCBC connections in the dorsal retina, in comparison to a prominent convergence of contacts from true S-cones to SCBCs in VN retina. Critically, the specificity of wiring from true S-cones to SCBCs also confirms the identity of true S-cones as revealed by opsin labeling and further supports the finding that true S-cones are highly concentrated in VN mouse retina.

## True S-cones in the ventral retina are not evenly distributed but form clusters

As demonstrated above, in the mouse retina, despite a large population of mixed cones, SCBCs precisely connect with true S-cones, preserving this fundamental mammalian color circuitry motif

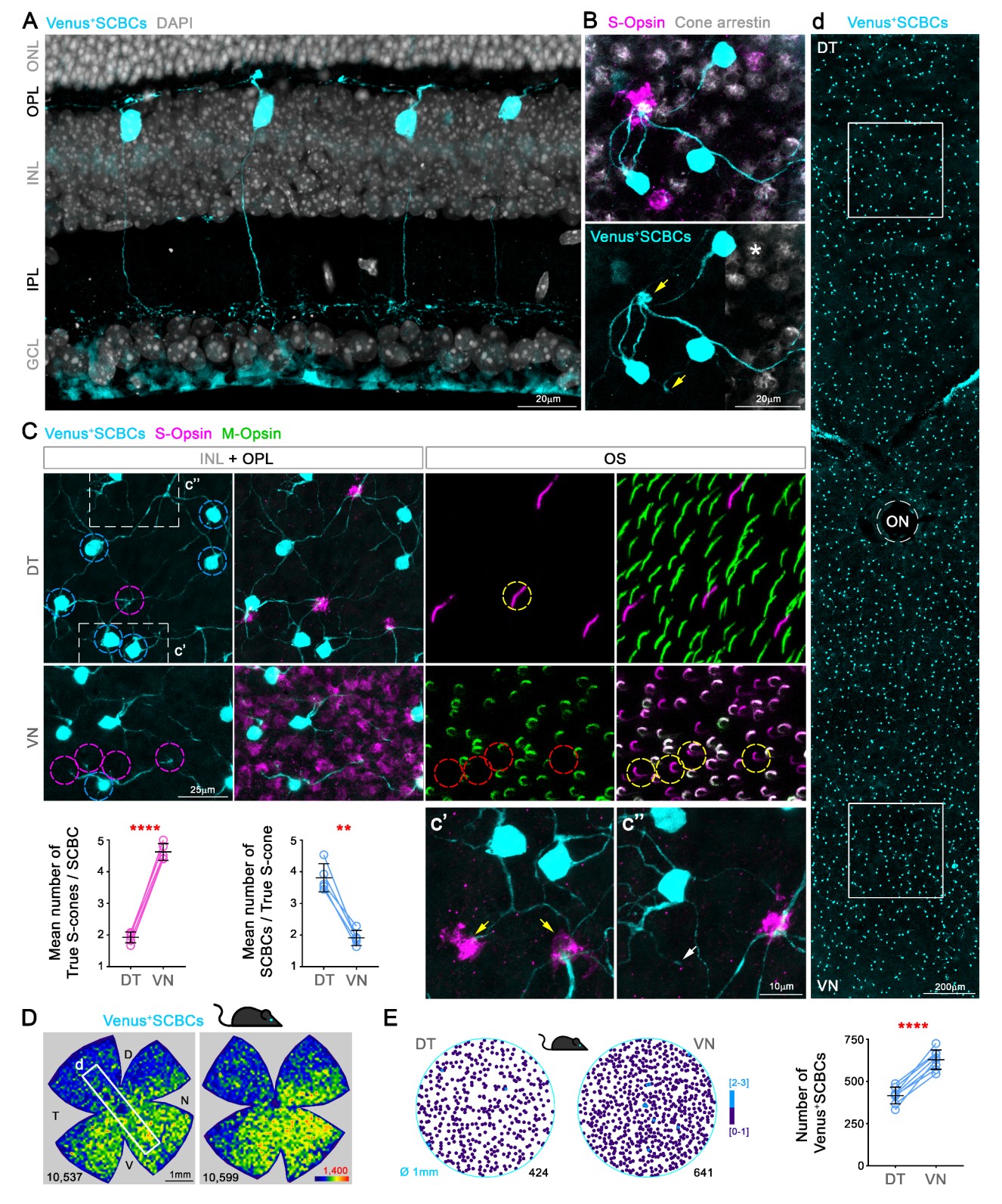

**Figure 3.** S-cone Bipolar cells (SCBCs) in Cpne9-Venus mouse retina. (**A**) Retinal cross section showing the characteristic morphology of SCBCs (***Behrens et al., 2016***; ***Breuninger et al., 2011***). (**B**) Detailed view of the selective connectivity between Venus⁺SCBCs and true S-cone terminals (yellow arrows). Note that SCBCs avoid contacts with cone terminals lacking S-opsin expression (M⁺S⁻-cone pedicles, identified using cone arrestin), as well as a mixed cone pedicle, marked with an asterisk. In fact, on the contrary, the SCBCs prefer to develop multiple contacts to the same true S-cone pedicle.

*Figure 3 continued on next page*

*Figure 3 continued*

(C) Images from flat-mount retinas focused on the inner nuclear and outer plexiform layers (INL+OPL) or in the photoreceptor outer segment (OS) layer of the corresponding area. Magnifications showing divergent and convergent connectivity patterns from true S-cone pedicles in dorsal and ventral retinal domains, respectively. In the DT retina, six Venus+ SCBCs (cyan circles) contact a single true S-cone pedicle (magenta circle in DT); while one Venus+ SCBC contacts at least four true S-cone pedicles in the VN retina (magenta circles in VN), which belong to cones possessing S+M-OSs (yellow circles). Connectivity between true S-cones and SCBCs in DT and VN retina was assessed as the average number of true S-cone pedicles contacting a single SCBC per retina (magenta plot) or the average number of SCBCs contacting a single true S-cone pedicle per retina (cyan plot) (p<0.0001, p<0.01, respectively; n = 5). (c') Detailed view of a secondary SCBC bifurcation contacting independently two true S-cone pedicles. (c") Detailed view of a 'blind' SCBC process. (D) Density maps depicting the distributions of SCBCs in Cpne9-Venus mice. (d) Venus+ SCBCs along the DT-VN axis from a flat-mount retina (corresponding to the white frame in D) showing the gradual increase of SCBCs towards the VN retina where true S-cone density peaks (last row in *Figure 2A*). (E) Demonstration of Venus+ SCBC densities color-coded by the k-nearest neighbor algorithm according to the number of other Venus+ SCBCs found within an 18 µm radius in two circular areas of interest (DT and VN). Although, Venus+ SCBCs exhibit a sparse density without forming clusters (circular maps), they were significantly denser in VN retina (p<0.0001; n = 8).

(*Behrens et al., 2016*; *Breuninger et al., 2011*; *Haverkamp et al., 2005*; *Mills et al., 2014*). However, the increased density of SCBCs in the ventral retina does not match that of true S-cones (compare *Figure 3D* and *Figure 2a*, last row). Thus, individual SCBCs in the ventral retina may be required to develop more dendrites to maximize the number of contacts made with different S-cone terminals (*Supplementary file 2*, graphs in *Figure 3C*). Intriguingly, we discovered in both strains that true S-cones in the ventral retina appeared to cluster together rather than forming an even distribution, as revealed by K-nearest neighbor analysis (*Figure 4A–B*, *Supplementary file 2*). Ideally, such true S-cone clustering may increase the availability of targets for individual SCBCs in a reduced space.

To quantify the spatial patterning of true S-cone populations (or their lack thereof), we compared the observed true S-cone distributions within 1 mm diameter VN and DT retinal samples to artificially generated alternative populations (*Figure 4C*). To this end, we considered two extreme patterning rules: First, one in which the space between true S-cone locations was maximized within the set of actual locations for all cones, creating a relatively uniform (evenly 'distributed') mosaic of true S-cones. At the other extreme, cone identities were permuted randomly ('shuffled') among observed cone locations (*Figure 4C*). Repetition of these algorithms generated distributions of patterning metrics for true S-cones (see below) that remain constrained by the observed cone locations and proportions of cone types for each 1 mm sample.

To quantitatively compare the patterning of real true S-cone populations to their artificial counterparts, we first computed two measures of regularity for true S-cones: nearest neighbor and Voronoi diagram regularity indices (NNRI and VDRI, respectively; *Reese and Keeley, 2015*; *Figure 4C–D*); larger values of these metrics indicate smaller variability in the spacing between cones and thus more regular patterns. Interestingly, far from being regularly distributed, true S-cone placement was quite irregular and nearly indistinguishable from shuffled populations (including a slight trend toward regularity measures lower than random, which may indicate a tendency toward clustering, *Figure 4D*; see *Reese, 2008*). To further probe the possibility of true S-cone clustering, we computed the ratios of true S-cone neighbors for each cone (denoted here as the S-cone neighbor ratio [SCNR]; see Methods for the calculation of the SCNR search radius for each retinal sample). Intriguingly, SCNRs were significantly larger for true S-cones than for other cone types, which were equal to expected ratios due to random chance—especially so in ventral retinas, further indicating a clustering of true S-cones in those areas (*Figure 4E*). Notably, a more extreme form of clustering of S-cones has been observed in the 'wild' mouse (*Warwick et al., 2018*) and with much lower densities in some felids (*Ahnelt et al., 2000*). Here, such clustering may reflect the mode of true S-cone development in the ventral retina, for example, by 'clonal expansion' to achieve unusually high densities (*Bruhn and Cepko, 1996*; *Reese et al., 1999*). It is tempting to speculate that it may also facilitate the wiring of true S-cones with sparsely distributed SCBCs, which were not observed to cluster in the ventral retina (*Figure 3E*). Indeed, we observed examples of groups of true S-cones forming clusters whose pedicles in the OPL were tightly congregated in a patch and contacted by a nearby SCBC (*Figure 4F*).

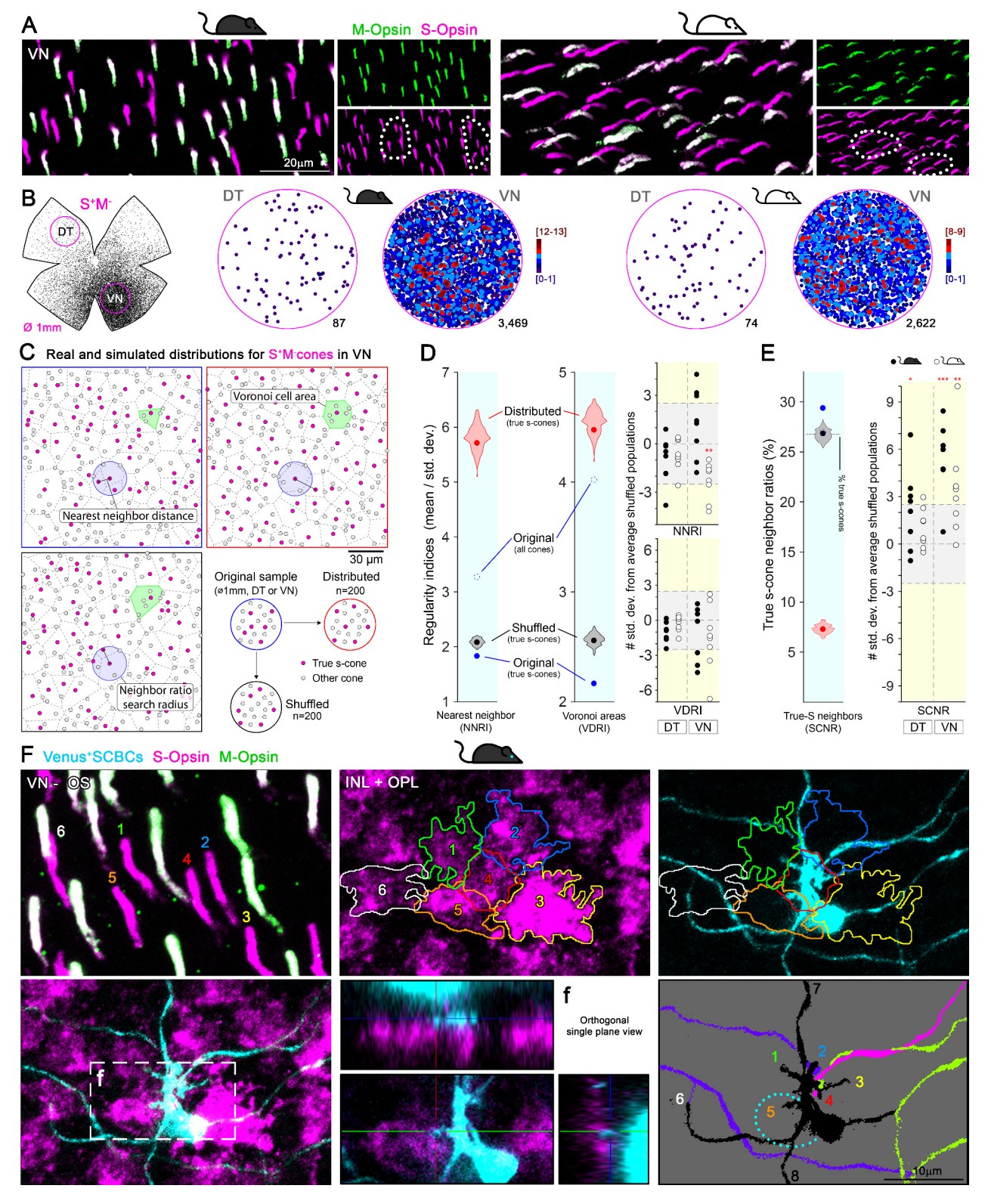

**Figure 4.** Clustering of true S-cones in the ventronasal (VN) retina. (**A**) Retinal magnifications from flat-mount retinas demonstrating grouping of true S-cones in the VN area, where true S-cone density peaks. White dashed lines depict independent groups of true S-cones that are not commingled with mixed cones (M$^+$S$^+$, white outer segments in the merged image). (**B**) Retinal scheme of true S-cones used for selecting two circular areas of interest along the dorsotemporal-ventronasal (DT-VN) axis. Circular maps demonstrate true S-cone clustering in these regions. True S-cone locations are color-

*Figure 4 continued on next page*

*Figure 4 continued*

coded by the k-nearest neighbor algorithm according to the number of other true S-cones found within an 18 µm radius. (C–E) Analytical comparisons of DT and VN populations of true S-cones to their simulated alternatives. (C) Example real and simulated true S-cone populations and their quantification. Images depict true S-cone locations (magenta dots) and boundaries of their Voronoi cells (dashed lines) from original and example simulated ('distributed', 'shuffled') cone populations. Gray dots indicate the locations of other cone types. Observed cone locations were used for all simulated populations; only their cone identities were changed. The annotated features are examples of those measurements used in the calculations presented in D-E. (D) Comparison of sample regularity indices for one albino VN retinal sample to violin plots of those values observed for n = 200 simulated cone populations. Note that average regularity indices for true S-cones were lower than that of shuffled populations, whereas those values lay between shuffled and distributed populations when all cones were considered. Plots on the right show values for all actual retinal samples normalized using the mean and standard deviations of their simulated 'shuffled' counterparts. The y-axis range corresponding to ±2.5 standard deviations from the mean (i.e., that containing ~99% of shuffled samples) is highlighted in gray. (E) Comparison of the real average SCNR for the example in C-D to those values for its simulated counterparts. Note that the average SCNR for all cones in this sample was equal to that predicted by random chance (i.e., the ratio of true S-cones to all cones), which in turn was equal to the average for true S-cones for shuffled samples. In contrast, the real true S-cone SCNR was higher. Plot on the right shows true S-cone SCNR values for all samples, normalized as described for D. (F) Convergent connectivity from a true S-cone cluster to a single SCBC in the VN retina. Images of a true S-cone cluster, in a flat-mount retina, focused on the photoreceptor outer segment layer and the inner nuclear-outer plexiform layers (INL+OPL). The upper left panel show the numerical and colored identification of each true S-outer segment in the cluster (note that the number positions indicate the locations where outer segments contact the photoreceptor inner segment). Each true S-cone pedicle belonging to this cluster is outlined and color coded (middle upper panel) and are overlaid upon the SCBC dendritic profile (right upper panel). To identify synaptic contacts between the SCBC and the cone pedicles (maximum intensity projection -excluding the SCBC soma- shown in lower left panel), we acquired orthogonal single plane views zooming into putative dendritic tips. An example for the contact with cone #5 is shown in lower middle panel, corresponding to the box area in lower left panel (f). The lower right panel shows dendritic endings of this SBCB (black) contacting the marked cones (#1–6). It also contacts two additional cones outside of the field of view (#7,8). Dashed line depicts the soma of the SCBC. Dendrites from other SCBCs are color coded for differentiation.

The online version of this article includes the following figure supplement(s) for figure 4:

**Figure supplement 1.** Reconstruction and mapping of true S-cone densities into visual space.

## Enriched true S-cones in the ventral retina may provide an anatomical base for mouse color vision

Despite being nocturnal and having a rod-dominated retina (*Carter-Dawson and LaVail, 1979*; *Jeon et al., 1998*), mice can detect color (*Denman et al., 2018*; *Jacobs et al., 2004*). Although it remains uncertain whether the source of long-wavelength sensitive signals for color opponency arises in rods or M-cones (*Baden and Osorio, 2019*; *Ekesten et al., 2000*; *Ekesten and Gouras, 2005*; *Joesch and Meister, 2016*; *Reitner et al., 1991*), it is clear that true S-cones provide short-wavelength signals for color discrimination. Given the previously-held notion that true S-cones are evenly distributed across the retina (*Baden et al., 2013*; *Franke et al., 2019*; *Haverkamp et al., 2005*; *Szatko et al., 2019*; *Wang et al., 2011*), whereas $M^+S^-$ cones are concentrated in the dorsal retina of pigmented mouse, it is intuitive to speculate that color coding is prevalent in the dorsal retina. However, previous physiological and behavioral studies indicate that, although luminance detection can occur across the mouse retina, color discrimination is restricted to the ventral retina (*Breuninger et al., 2011*; *Denman et al., 2018*; *Szatko et al., 2019*). Thus, our discovery of high enrichment of true S-cones in the ventral retina provides a previously missed anatomical feature for mouse color vision that could help to re-interpret these results. From projections mapping true S-cone densities into visual space (*Figure 4—figure supplement 1*; *Sterratt et al., 2013*), it is conceivable that high ventral true S-cone density will provide a much higher sensitivity of short-wavelength signals, thus facilitating color detection for the upper visual field. Although the true S-cone signals carried by SCBCs in the dorsal retina might not be significant for color detection, they could certainly participate in other functions, such as non-image forming vision, that are known to involve short-wavelength signals (*Altimus et al., 2008*; *Doyle et al., 2008*; *Patterson et al., 2020*). Interestingly, the overall true S-cone percentage in the mouse retina remains approximately 10% (*Figure 2B*), and the average true S-cone to SCBC ratio across the whole retina is about 1.7:1 (*Supplementary file 1B-C*), similar to what has been reported in other mammals (*Ahnelt et al., 2006*; *Ahnelt and Kolb, 2000*; *Bumsted et al., 1997*; *Bumsted and Hendrickson, 1999*; *Curcio et al., 1991*; *Hendrickson and Hicks, 2002*; *Hunt and Peichl, 2014*; *Kryger et al., 1998*; *Lukáts et al., 2005*; *Müller and Peichl, 1989*; *Ortín-Martínez et al., 2010*; *Peichl et al., 2000*; *Schiviz et al., 2008*; *Shinozaki et al., 2010*; *Szél et al., 1988*).

Such a spatial rearrangement of true S-cones and SCBCs likely reflects evolutionary adaptation to enhance short-wavelength signaling and color coding for the upper visual field as best suited for the habitat and behavior of mice (*Baden et al., 2020*). For example, it may facilitate aerial predator detection during daytime (*Yilmaz and Meister, 2013*). Similarly, skewed S-cone arrangement has been reported for other terrestrial prey mammals (*Famiglietti and Sharpe, 1995*; *Juliusson et al., 1994*; *Röhlich et al., 1994*), while zebrafish possess a UV-enriched ventral retina that enhances their predation (*Zimmermann et al., 2018*). In addition, we observed that the clustering of true S-cones in the ventral retina may allow several neighboring cones of the same type to converge onto the same SCBC (*Figure 4F*), which could potentially enhance signal-to-noise ratios for more accurate detection, as described recently in human fovea (*Schmidt et al., 2019*). It is also remarkable that despite the very different S-opsin expression patterns in both mouse strains, the true S-cone population and distribution are strikingly similar between pigmented and albino mice, suggesting a common functional significance.

# Materials and methods

## Key resources table

| Reagent type (species) or resource | Designation | Source or reference | Identifiers | Additional information |
|---|---|---|---|---|
| Strain, strain background (*Mus musculus*, male) | C57BL/6J mouse strain | Jackson Laboratory | Cat#000664, RRID:IMSR_JAX:000664 | Pigmented mouse inbred strain |
| Strain, strain background (*Mus musculus*, male) | Crl:CD-1(ICR) mouse strain | Charles River | Cat#022, RRID:IMSR_CRL:022 | Albino mouse strain |
| Strain, strain background (*Mus musculus*, male) | Copine9-Venus mouse line | This paper | | Material and methods section 8.3.1 |
| Antibody | anti-OPN1SW (N-20) (Goat polyclonal) | Santa Cruz Biotechnology | Cat#sc-14363, RRID:AB_2158332 | IF (1:1200) |
| Antibody | anti-Opsin Red/Green (Rabbit polyclonal) | Millipore/Sigma | Cat#AB5405, RRID:AB_177456 | IF (1:1000) |
| Antibody | anti-Cone Arrestin (Rabbit polyclonal) | Millipore/Sigma | Cat#AB15282, RRID:AB_1163387 | IF (1:300) |
| Antibody | anti-GFP (Chicken polyclonal) | Millipore/Sigma | Cat#AB16901, RRID:AB_11212200 | IF (1:100) |
| Antibody | anti-Rabbit 488 (Donkey polyclonal) | Jackson Immunoresearch | Cat#711-547-003, RRID:AB_2340620 | IF (1:500) |
| Antibody | anti-Rabbit Cy3 (Donkey polyclonal) | Jackson Immunoresearch | Cat#711-165-152, RRID:AB_2307443 | IF (1:500) |
| Antibody | anti-Goat 647 (Donkey polyclonal) | Jackson Immunoresearch | Cat#705-605-147, RRID:AB_2340437 | IF (1:500) |
| Antibody | anti-Goat Cy3 (Donkey polyclonal) | Jackson Immunoresearch | Cat#705-166-147, RRID:AB_2340413 | IF (1:500) |
| Antibody | anti-Chicken 488 (Donkey polyclonal) | Jackson Immunoresearch | Cat#703-545-155, RRID:AB_2340375 | IF (1:500) |

*Continued on next page*

*Continued*

| Reagent type (species) or resource | Designation | Source or reference | Identifiers | Additional information |
|---|---|---|---|---|
| Sequence-based reagent | Copine9_gRNA_L (73/25) | This paper | | 5'GAGACATGACTGGTCCAA3' |
| Sequence-based reagent | Copine9_gRNA_R (62/4.40), | This paper | | 5'GCCTCGGAGCGTAGCGTCC3' |
| Software, algorithm | Zen | Zeiss | Zen lite Black edition 2.3 SP1 | |
| Software, algorithm | FIJI-ImageJ | NIH | v1.52r | https://imagej.nih.gov/ij/ |
| Software, algorithm | Sigma Plot | Systat Software | 13.0 | |
| Software, algorithm | GraphPad Prism | Graph Pad Software | 8.3.0 | |
| Software, algorithm | Photoshop | Adobe | CC 20.0.6 | |
| Software, algorithm | MATLAB | MathWorks | 2016 | |
| Software, algorithm | R | The R Project for Statistical Computing | 3.5.3 | https://www.r-project.org/ |
| Software, algorithm | Retina and Visual Space Retistruct Package | Sterratt DC et al., PLoS Comput Biol. | | |
| Software, algorithm | Zotero | Corporation for Digital Scholarship | 5.0 | https://www.zotero.org/download/ |
| Other | DAPI | ThermoFisher Scientific | Cat# D3571, RRID:AB_2307445 | (1 ug/ml) |

## Animal generation, handling and ethic statement

Three months old male pigmented (C57BL/6J, n = 5), albino (CD1, n = 5) mice were obtained from the National Eye Institute breeding colony. The Venus-Cpne9 mouse line (n = 5; based on previous single cell sequencing data [*Shekhar et al., 2016*]) carries a reporter (Venus) allele under the control of the mouse Cpne9 locus. The reporter allele was created directly in B6.SJL(F1) zygotes using CRISPR-mediated homologous recombination (HR) (*Yang et al., 2013*). Briefly, a HR targeting template was assembled with PCR fragments of 5' and 3' homology arms of 910 bp and 969 bp respectively, flanking exon one, and a Venus expression cassette carrying the bovine growth hormone polyadenylation (bGH-PolyA) signal sequence as the terminator. Homology arms were designed such that integration of the reporter cassette would be at the position right after the first codon of the Cpne9 gene in exon one. A pair of guide RNAs (gRNA), with outward orientation (38 bp apart), were synthesized by in vitro transcription as described (*Yang et al., 2013*) and tested for their efficiency and potential toxicity in a zygote differentiation assay where mouse fertilized eggs were electroporated with SpCas9 protein and gRNA ribonuclear particles. Eggs were cultured in vitro for 4 days in KSOM (Origio Inc, CT) until differentiated to blastocysts. Viability and indel formation were counted respectively. gRNA sequences are (1) Copine9_gRNA_L(73/25), 5'GAGACATGACTGGTCCAA3'; (2) Copine9_gRNA_R(62/4.40), 5'GCCTCGGAGCGTAGCGTCC3'. A mixture of the targeting plasmid (super coiled, 25 ng/μl) with two tested gRNAs (25 ng/μl each) and the SpCas9 protein (Life Science technology, 30 ng/μl) were microinjected into mouse fertilized eggs and transferred to pseudopregnant female recipients as described elsewhere (*Yang et al., 2013*). With a total of 15 F0 live births from 6 pseudopregnant females, 11 were found to carry the knockin allele by homologous recombination, a HR rate of 73%. F0 founders in B6.SJL F1 (50% C57BL6 genome)

were crossed consecutively for 3 generations with C57BL6/J mice to reach near congenic state to C57BL6/J.

Mice were housed a 12:12 hr light/dark cycle. All experiments and animal care are conducted in accordance with protocols approved by the Animal Care and Use Committee of the National Institutes of Health and following the Association for Research in Vision and Ophthalmology guidelines for the use of animals in research.

### Tissue collection

All animals were sacrificed with an overdose of $CO_2$ and perfused transcardially with saline followed by 4% paraformaldehyde. To preserve retinal orientation, eight retinas per mouse strain/line were dissected as flat whole-mounts by making four radial cuts (the deepest one in the dorsal pole previously marked with a burn signal as described [*Nadal-Nicolás et al., 2018*; *Stabio et al., 2018b*]). The two remaining retinas were cut in dorso-ventral orientation (14 µm) after cryoprotection in increasing gradients of sucrose (Sigma-Aldrich SL) and embedding in optimal cutting temperature (OCT; Sakura Finetek).

### Immunohistochemical labeling

Immunodetection of flat-mounted retinas or retinal sections was carried out as previously described (*Nadal-Nicolás et al., 2018*). Importantly, the retinal pigmented epithelium was removed before the immunodetection. First, whole-retinas were permeated (4 × 10') in PBS 0.5% Triton X-100 (Tx) and incubated by shaking overnight at room temperature with S-opsin (1:1200) and M-opsin (1:1000) or cone arrestin (1:300) primary antibodies diluted in blocking buffer (2% normal donkey serum). Cpne9-Venus retinas were additionally incubated with an anti-GFP antibody (1:100) to enhance the original Venus signal. Retinas were washed in PBS 0.5% Tx before incubating the appropriate secondary antibodies overnight (1:500). Finally, retinas were thoroughly washed prior to mounting with photoreceptor side up on slides and covered with anti-fading solution. Retinal sections were counterstained with DAPI.

### Image acquisition

Retinal whole-mounts were imaged with a 20x objective using a LSM 780 Zeiss confocal microscope equipped with computer-driven motorized stage controlled by Zen Lite software (Black edition, Zeiss). M- and S-opsins were imaged together to allow the identification and quantification of different cone types. Magnifications from flat mounts and retinal cross-sections (*Figure 1*) were taken from dorsal, medial and ventral areas using a 63x objective for opsin co-expression analysis. Images from retinal cross-sections were acquired ~1.5 mm dorsally or ventrally from the optic disc.

### Sampling and opsin co-expression measurement

In four retinas per strain, we acquired images from three 135 × 135 µm samples (63x) per each area of interest (dorsal, medial and ventral). These areas were selected according to the S-opsin gradient in wholemount retinas (see scheme in *Figure 1C*). Cone outer segments were manually classified as $M^+S^-$, true S- ($S^+M^-$) or mixed ($M^+S^+$) cones depending on their opsin expression. Data representation was performed using GraphPad Prism 8.3 software.

### Image processing: manual and automated whole quantification

To characterize the distribution of the different cone photoreceptor types in the mouse retina, we developed and validated an automatic routine (ImageJ, NIH) to identify, quantify the total number of outer segments and finally extract the location of each individual cone (*Figure 2—figure supplement 1A*). Briefly, maximum-projection images were background-subtracted and thresholded (background-noise mean value, 9.6 ± 1.2% and 15.2 ± 3.2% for S- and M-opsin respectively, the threshold was applied at 15.7%) to create a binary mask that was then processed using watershed and despeckle filters to isolate individual cones and reduce noise. The '3D Objects Counter' plugin was applied to such images to count cones within fixed parameters (shape and size) and extract their *xy* coordinates for further analysis. This automation was validated by statistical comparison with manual counting performed by an experienced investigator (Pearson correlation coefficient $R^2$ = 96–99% for M- or S-opsin respectively, *Figure 2—figure supplement 1B*). To count cone subtypes, images were

pre-processed with image processing software (Adobe Photoshop CC) to isolate the desired subtype and then manually marked using Photoshop, or automatically counted using ImageJ as described above. Total cone populations were determined by combining M- and S-opsin channels, while mixed $M^+S^+$ cones were obtained by masking the M-opsin signal with the S-opsin channel. $M^+S^-$ cones in pigmented mice were obtained by subtracting the S-opsin signal from the M-opsin photomontage. Finally, $M^+S^-$ cones (in albino samples), true S-cones (both strains) (**_Figure 2—figure supplement 1C_**) and Venus$^+$ SCBCs (Cpne9-Venus mouse line) were manually marked on the retinal photomontage (Adobe Photoshop CC).

## Topographical distributions

Topographical distributions of cone population densities were calculated from cone locations identified in whole-mount retinas using image processing (see above). From these populations, isodensity maps were created using Sigmaplot 13.0 (Systat Software). These maps are filled contour plots generated by assigning to each area of interest (83.3 × 83.3 μm) a color code according to its cone density, ranging from 0 (purple) to 17,300 cones/mm$^2$ for all cone types except for true S-cones and $M^+S^-$-cone in the albino strain (5000 cones/mm$^2$), as represented in the last image of each row of **_Figure 2A_**, or 1,400 SCBCs/mm$^2$ (**_Figure 3D_**) within a 10-step color-scale. These calculations allow as well, the illustration of the number of cones at a given position from the ON center. To analyze the relative opsin expression along the retinal surface, we have considered three cone populations (mixed, $M^+S^-$ and true S-cones) dividing the retina in four quadrants: dorsotemporal, dorsonasal, ventrotemporal and ventronasal (DT, DN, VT and VN respectively, scheme in **_Figure 2C_**). The relative percentage of cone-types is represented in line graphs from four retinas/strain (SigmaPlot 13.0).

## SCBC sampling and 'true S-cone' connectivity

To characterize the connectivity of Venus$^+$ S-cone bipolar cells (Venus$^+$ SCBCs) with true S-cone terminals, we acquired images from the same area (260 × 260 μm, 63x) at two focal planes: First, we focused upon the INL+OPL, then the corresponding photoreceptor outer segment (OS) layer, respectively, for two areas of interest (DT and VN). To verify connectivity between Venus$^+$ SCBC dendrites and true S-cone pedicles in the OPL, in addition to S-opsin immunodetection, we also labeled retinas using cone arrestin antibodies to discriminate mixed cone pedicles from true S-cone pedicles, because true S-cone pedicles contain either low or no cone arrestin (**_Figure 3B_**, **_Haverkamp et al., 2005_**). In other retinas, SCBC contacts were verified by tracking each cell body from cone pedicles to their respective OS to confirm S$^+$M$^-$ opsin labeling (**_Figure 3C_**). In five retinas (with S- and M-opsin double immunodetection), we analyzed the connectivity between 186 Venus$^+$ SCBCs (133 and 53 for DT and VN respectively) and 263 true S-cone pedicles (74 and 189, DT and VN respectively). The number of synaptic contacts was assessed by tracking manually each SCBC-branch from the cell body using the Zen lite black visualization package (Z-stack with 1 μm interval). Multiple branch contacts in one true S-cone pedicle from a single SCBC were considered a single contact and counted only once (**_Figure 3B_**), while secondary bifurcations were considered as multiple contacts (**_Figure 3c'_**). SCBC-blind endings were not counted (**_Figure 3c''_**). The average number of contacts per retina was used to calculate the DT and VN means (**_Supplementary file 2_** and graphs in 3C).

## Clustering analysis

### K-neighbor maps and variance analysis of Voronoi dispersion

To assess the true S-cones and S-cone bipolar cell (SCBC) clustering, we performed two comparable sets of analyses. First, we extracted two circular areas (1 mm diameter) in the DT-VN axis at 1 mm from the optic disc center (scheme in 4B). A K-nearest neighbor algorithm (**_Nadal-Nicolás et al., 2014_**) was used to map the number of neighboring true S-cones within a 18 μm radius of each true S-cone to a color-code in its retinal position (**_Figure 4B_**). Regularity indices were computed for each retinal sample using Voronoi diagrams for cone positions as well as nearest neighbor distances (VDRI and NNRI, respectively [**_Reese and Keeley, 2015_**; **_Figure 4C–E_**]). NNRIs were computed as the ratio of the mean to the standard deviation for the distance from true S-cones to their nearest true S-cone neighbor. true S-cone neighbor ratios (SCNR) were calculated for each retinal sample as the average proportion of true s-cones within a given radius for each cone. This search radius was

calculated separately for each sample to correct for sample-to-sample variations in total density: this radius ($r$) was calculated as $r = 3\sqrt{(A / (\sqrt{2}\,\pi N))}$, where $A$ is the circular area of the 1 mm diameter retinal sample and $N$ is the total number of cones in that sample. For a highly regular cell mosaic containing $N$ cells filling an area $A$, this calculation estimates the location of the first minimum in the density recovery profile (*Rodieck, 1991*), providing the average radius of a circle centered upon a cone that will encompass its first tier of cone neighbors (but exclude the second tier) in an evenly distributed mosaic. To minimize edge effects from computations of NNRI, VDRI, SCNR, those values for cones closer to the outer edge of the sample than the SCNR search radius were discarded. To produce simulated cone mosaics for comparison with observed values, cone distributions with evenly 'distributed' true S-cones were generated by first using a simple mutual repulsion simulation to maximize the distances between true S-cones, followed by assigning the nearest positions among all cone locations as being 'true S'. 'Shuffled' populations of true S-cones were generated by permuting cone identities randomly among all cone locations, holding the proportion of true S-cones constant. Voronoi diagrams, neighbor calculations, and mosaic generation and other computations were performed using MATLAB R2016b.

### True S-cone cluster and SCBC synaptic contacts evaluation

To characterize the true S-cone cluster connectivity in the VN retina, retinal whole-mounts were imaged with a 63x objective, from the photoreceptor outer segments to the OPL, in a Z-stack image with 0.5 µm interval. To visualize the true S-cone clustering and Venus$^+$ SCBC connectivity, we identified numerically, and color coded each true S-outer segment form a cluster. The corresponding true S-pedicles were identified by tracking the cell body from their S$^+$M$^-$OSs. Focusing on the outer plexiform layer (OPL), each individual true S-cone pedicle -that form a cluster- was manually outlined and color coded accordingly. Lastly, the SCBC synaptic terminals that belong to a single SCBC, were identified by their specific contacts to the respective true S-cone pedicle (*Figure 4F*).

### Retinal reconstruction and visuotopic projection

Retinal images were reconstructed and projected into visual space using R software v.3.5.2 for 64-bit Microsoft Windows using Retistruct v.0.6.2 as in *Sterratt et al., 2013*. Reconstruction parameters from that study were used: namely, a rim angle of 112° (phi$_0$ = 22°), and eye orientation angles of 22° (elevation) and 64° (azimuthal). For *Figure 4—figure supplement 1*, true S-cone density contour lines and heatmaps were computed in MATLAB and overlaid onto flat-mount retina opsin labeling images using ImageJ prior to processing by Retistruct.

### Statistical analysis

Statistical comparisons for the percentage of cones/retinal location, the total cone quantifications (*Supplementary file 1*) and the DT or VN true S-cones and Venus$^+$SCBCs (*Supplementary file 2*) were carried out using GraphPad Prism v8.3 for Microsoft Windows. Data are presented as mean ± standard deviation. All data sets passed the D'Agostino-Pearson test for normality, and the comparisons between strains were performed with Student's *t*-test.

For each 1 mm retinal sample, VDRI, NNRI, and SCNR values were normalized and compared to the distributions of 'shuffled' cone populations. Such comparisons were not performed against 'distributed' populations, because in those populations, VDRI and NNRI values were consistently much higher—and SCNR much lower—than in real samples (see *Figure 4D–E*). The 'shuffled' populations for each retinal region produced measurements that were well described by normal distributions (Kolmogorov-Smirnov test, MATLAB). Thus, to allow comparisons across samples, we converted each measurement into a Z-score using the mean and standard deviation of those measures from shuffled populations. One-tailed Student's *t*-tests were performed to compare the normalized measures to the distribution of 'randomly shuffled' cone population measures, and significance was determined at the p<0.05 level.

## Acknowledgements

The authors would like to thank the NEI Animal Care team, especially Megan Kopera and Ashley Yedlicka.

## Additional information

### Funding

| Funder | Grant reference number | Author |
| --- | --- | --- |
| National Eye Institute | Intramural Research Program | Wei Li |

The funders had no role in study design, data collection and interpretation, or the decision to submit the work for publication.

### Author contributions

Francisco M Nadal-Nicolás, Conceptualization, Data curation, Software, Formal analysis, Supervision, Validation, Investigation, Visualization, Methodology, Writing - original draft, Writing - review and editing; Vincent P Kunze, Conceptualization, Writing - review and editing; John M Ball, Data curation, Software, Formal analysis, Validation, Visualization, Writing - review and editing; Brian T Peng, Akshay Krishnan, Gaohui Zhou, Formal analysis, Investigation, Methodology, Writing - review and editing; Lijin Dong, Resources, Methodology, Writing - review and editing; Wei Li, Conceptualization, Resources, Supervision, Funding acquisition, Visualization, Writing - original draft, Project administration, Writing - review and editing

### Author ORCIDs

Francisco M Nadal-Nicolás (iD) https://orcid.org/0000-0003-4121-514X
Vincent P Kunze (iD) https://orcid.org/0000-0002-7869-9793
Wei Li (iD) https://orcid.org/0000-0002-2897-649X

### Ethics

Animal experimentation: All experiments and animal care are conducted in accordance with protocols approved by the Animal Care and Use Committee of the National Institutes of Health and following the Association for Research in Vision and Ophthalmology guidelines for the use of animals in research. The protocol was approved by the Animal Care and Use Committee of the National Institutes of Health (ASP#606).

### Decision letter and Author response

Decision letter https://doi.org/10.7554/eLife.56840.sa1
Author response https://doi.org/10.7554/eLife.56840.sa2

## Additional files

### Supplementary files

• Source data 1. Raw quantitative data and statistics analysis.

• Supplementary file 1. Quantifications of cone-type populations and S-cone bipolar cells. (**A**) Cone numbers in different retinal areas along the dorsoventral axis in pigmented and albino mouse. Three images/area (dorsal, medial and ventral) from four retinas/strain. Different cone type quantifications are shown as average ± SD, corresponding to the percentages shown in *Figure 1C*. The total number of cones analyzed per location and strain are shown in the last column. Total number of cones (**B**) or S-cone Bipolar cells (SCBCs, **C**) in eight retinas/mouse strain or line (average ± SD, see also *Figure 2B*). Significant differences between strains $p<0.05$ (*), $p<0.01$ (**), $p<0.001$ (***), $p<0.0001$ (****).

• Supplementary file 2. True S-cone terminals and Cpne9-Venus+SCBCs connectivity in dorsotemporal (DT) and ventronasal retina (VN). Quantitative data are shown as mean ± SD from the average of five DT and VN retinal areas (*Figure 3C*). Significant differences between retinal areas, $p<0.01$ (**), $p<0.0001$ (****).

• Supplementary file 3. True S-cones and S-cone bipolar cells in dorsotemporal and ventronasal retinas. Numbers of true S-cones (**A**) and Cpne9-Venus$^+$SCBCs (**B**) in dorsotemporal (DT) and ventronasal (VN) circular areas (1 mm diameter, *Figures 3E* and *4B*). Quantitative data are shown as average ± SD from eight retinas/strain or line. The mean of true S-cones and Venus$^+$SCBCs in these circular areas was used to calculate the DT:VN and true S-cone:SCBC (**C**) ratios. Significant differences between strains p<0.05 (*), p<0.001 (***). True S-cones and SCBCs were significant different between DT and VN retina (p<0.0001).

• Transparent reporting form

### Data availability

All data generated or analyzed during this study are included in the manuscript and supporting files.

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
