## [Decision Letter]

**Acceptance summary:**

This paper shows that true S cones in mouse retina are highly concentrated in the ventral retina. This is a substantial departure from the current picture which holds that S cones are uniformly distributed across the retina, and the present results will require reevaluating models for chromatic sensitivity in mice.

**Decision letter after peer review:**

Thank you for submitting your article "True S-cones are concentrated in the ventral mouse retina for color detection in the upper visual field" for consideration by *eLife*. Your article has been reviewed by three peer reviewers, including Fred Rieke as the Reviewing Editor and Reviewer #1, and the evaluation has been overseen Barbara Shinn-Cunningham as the Senior Editor The following individuals involved in review of your submission have agreed to reveal their identity: Tom Baden (Reviewer #2); Raunak Sinha (Reviewer #3).

The reviewers have discussed the reviews with one another and the Reviewing Editor has drafted this decision to help you prepare a revised submission.

We would like to draw your attention to changes in our revision policy that we have made in response to COVID-19 (https://elifesciences.org/articles/57162). Specifically, we are asking editors to accept without delay manuscripts, like yours, that they judge can stand as *eLife* papers without additional data, even if they feel that they would make the manuscript stronger. Thus the revisions requested below only address clarity and presentation. There are (optional) comments about further experiments, which we have retained so that you can see those suggestions but which are not required.

The reviewers were all in agreement that the paper provides important new information about cone types in mouse retina, and that the paper is in good shape. Comments in the individual reviews (below) highlight some points that could be strengthened.

Reviewer #1:

The sampling of visual space by different cone photoreceptor types is an important determinant of how color vision works. Mice have long been known to show a dorsal-ventral gradient in the relative expression of the S and M cone opsin. Many mouse cones express both opsins, with a ratio in expression level of cones in different parts of the retina reflecting the dorsal-ventral gradient. In addition to these "mixed" cones, mice also have true S cones – cones that only express the S cone opsin. Those cones, and their distribution across the retina, are the topic of this paper.

The authors find that true S cones are highly concentrated in the ventral retina. This is based on direct labeling of the cone opsin, and also by generation of a mouse with a class of bipolar cells that exclusively contact S cones labeled. This conclusion is well supported by the data shown, and provides interesting and new information about the location of S cones in mouse retina. A final conclusion – that the true S cones form clusters – could be presented more clearly, as detailed below along with several other concerns/suggestions.

S cone clustering

One of the highlighted results is that the true S cones form clusters, and that this may lessen the required length of the dendrites of the S cone bipolar cells. Clustering to me implies a non-random distribution, and indeed the data provide support for this. But much stronger is the evidence that the distribution is not regular. The text on this point is confusing, and I think it would help to set up the issue more clearly from the start by introducing the range of possibilities (i.e. from a deterministic mosaic, to random placement).

Related to this point is the relation between cone locations and bipolar dendritic length. It would be helpful to generate models for the required bipolar dendritic length given the number of cone contacts for the three cone distributions and see what those predict. At present, given the apparent lack of evidence for clustering, it is hard to evaluate the suggestion about wiring length.

Presentation

The text in general could be expanded to help guide a reader through the figures. I found myself reading portions of the text and figure legends multiple times before I could understand the content of a given figure, and I am not sure I would have had that patience had I not been reviewing the paper. The results are clear and quite beautiful, so guiding someone through them clearly is important! As a general guide I think the paper could expand in length by a factor of ~2.

Here are some examples of places I found myself spending a lot of time to understand. (1) Figure 1C. Visually I expected the outer ring to follow from the inner pie chart, and it took a while to realize what was plotted there. (2) Figure 2C. More description in the text of what is plotted here would help. Maybe pick a quadrant and describe each line carefully. (3) A more complete description of how different cone terminals were identified for Figure 3 would be helpful. This is not a complete list – so I would ask the authors to go through the paper and make sure they lead the reader through each figure carefully and not just give the result.

Quantification

In general, the results are nicely described and then quantified. Figure 3 is an exception, where there are several statements made without any quantitative analysis. It is important, for example, to see some numbers about the types of cone contacts made by the S cone bipolar cells. Similarly, more complete numbers for both convergence and divergence in dorsal and ventral retina are needed (fleshing out details in subsection “2.3. S-cone bipolar cells exhibit a dorsal-ventral gradient with a higher density in the ventral retina.”).

Reviewer #2:

Nadal-Nicolas et al., present an elegant and important survey of mouse cone distributions across the retina. They show that many S-opsin expressing cones in the ventral retina are in fact "true" S-cones, unlike previously thought. This has important consequences for retinal wiring, colour vision and our understanding of ecological adaptation in vision.

I had commented on an earlier version of this work elsewhere. At the time I thought it was excellent and certainly worth publishing in a broad journal, and I continue to think so also now. From that previous version, the authors suitably addressed most of my points raised at the time, accordingly all my remaining points on this new and improved version are minor.

Reviewer #3:

In this study, Nadal-Nicolas and colleagues have used immunohistochemistry and careful anatomical analysis to identify an interesting distribution of cone photoreceptors, that only express short wavelength-sensitive opsin pigment (true S-cones), in mouse retina. Most cones in mouse retina express both the medium wavelength (M) and short-wavelength (S) sensitive opsin. It is well established that there is spatial gradient of opsin expression across the mouse retina – with middle wavelength-sensitive and S opsin enriched in the dorsal and ventral retina respectively. However, previous studies have proposed a uniform distribution of true S-cones across the dorso-ventral axis. This issue has been revisited by Nadal-Nicolas and colleagues where they systematically map the distribution of all cone types i.e. M^+^S^+^, M^+^S^-^ and S^+^M^-^, and surprisingly find that the concentration of true S-cones (S^+^M^-^) is remarkably higher in the ventral retina than in the dorsal retina. When comparing this cone distribution with albino mouse, they find that even though there is robust S opsin expression in the dorsal retina (unlike pigmented mouse), the pattern of true S cone distribution remained mostly unchanged. The authors further tested if the bipolar cells that receive synaptic inputs from the S cones follow the uneven distribution of the true S-cones. They convincingly demonstrate using a novel transgenic mouse, which selectively labels the S-cone bipolar cells, that these bipolar cells mimic the dorsal ventral concentration gradient as the true-S cones. These findings are novel and important for a mechanistic understanding of color vision. The data quality is high and the analysis is robust. Overall, it is well-laid out study with beautiful figures and exciting results.

I have a few comments that could potentially add to their findings:

i) Is there divergent contacts from a single true-S cone to SCBCs in the ventral retina like in the dorsal retina? Also, it would be informative to know if the SCBCs also get input from the M^+^S^+^ cones compared to S^+^M^-^ cones. How does this change going between dorsal, medial and ventral retina? This is important for making the argument about sensitivity of color opponent signals being higher in the ventral retina.

ii) The authors have used a robust automated and a manual analysis method to identify and count each of the cone types. In subsection “10.3.6. Image processing: manual and automated whole quantification”, the authors mention using a threshold to create a binary mask and subsequent filtering. It would be great to mention in the methods where the threshold is placed relative to the background noise in terms of standard deviations above noise. Also, could the placement of the threshold be a reason for mis-classifying a small fraction of M^+^S^+^ cones as true S cones since M opsin expression is low?

iii) It is indeed interesting to see the clustered arrangement of the true-S cones in the ventral retina. Is this also a feature of the true M cones (M^+^S^-^) in the dorsal retina? Moreover, is it possible to compare a M cone contacting bipolar cell (any type) and if it has preference of contacting the true M cones vs the M^+^S^+^ cones and if this changes between dorsal and ventral retina. The authors have a great opportunity to also expand on the true M cone distribution and its connectivity across mouse retina.

iv) The authors have generated a SCBC specific mouse transgenic line which could be very useful for targeting this bipolar cell type for single cell electrophysiology to substantiate the anatomical findings in this study. In particular, the divergent vs convergent inputs from true S cones to SCBCs in dorsal vs ventral retina should have a direct bearing on the strength of the synaptic inputs on SCBCs and this could be validated by measuring short-wavelength light evoked excitatory currents from SCBCs.

---

## [Author Response]

Reviewer #1:The sampling of visual space by different cone photoreceptor types is an important determinant of how color vision works. Mice have long been known to show a dorsal-ventral gradient in the relative expression of the S and M cone opsin. Many mouse cones express both opsins, with a ratio in expression level of cones in different parts of the retina reflecting the dorsal-ventral gradient. In addition to these "mixed" cones, mice also have true S cones – cones that only express the S cone opsin. Those cones, and their distribution across the retina, are the topic of this paper.The authors find that true S cones are highly concentrated in the ventral retina. This is based on direct labeling of the cone opsin, and also by generation of a mouse with a class of bipolar cells that exclusively contact S cones labeled. This conclusion is well supported by the data shown, and provides interesting and new information about the location of S cones in mouse retina. A final conclusion – that the true S cones form clusters – could be presented more clearly, as detailed below along with several other concerns/suggestions.S cone clusteringOne of the highlighted results is that the true S cones form clusters, and that this may lessen the required length of the dendrites of the S cone bipolar cells. Clustering to me implies a non-random distribution, and indeed the data provide support for this. But much stronger is the evidence that the distribution is not regular. The text on this point is confusing, and I think it would help to set up the issue more clearly from the start by introducing the range of possibilities (i.e. from a deterministic mosaic, to random placement).

We have carefully edited that section, after a brief review of the current knowledge on true S-cones and S-cone bipolar cells distribution in the retina, we have included a paragraph to concisely explain the purpose of generating different simulated cone populations and to guide the readers through the corresponding results and figures.

Related to this point is the relation between cone locations and bipolar dendritic length. It would be helpful to generate models for the required bipolar dendritic length given the number of cone contacts for the three cone distributions and see what those predict. At present, given the apparent lack of evidence for clustering, it is hard to evaluate the suggestion about wiring length.

Thank you for the insightful suggestion. We have classified cone populations here using opsin expression, which clearly labels outer segments, but for technical reasons, m-opsin antibodies very poorly penetrate to synaptic terminals, preventing reliable classification of cone pedicles. In addition, we do not yet have true S-cone and SCBC maps from the same retina, which would be the most useful information for making such estimates. Currently, we are in the process of crossing the SCBC line with an S-opsin reporter line and with a trb2-GFP line, which will make it possible to better visualize and quantify cone pedicle distribution of different types as well as SCBC distributions. It will definitely be valuable to revisit this topic once these materials are available. Nonetheless, we did perform some additional analysis in retinal areas that allowed us to trace S-opsin labeling to the pedicles, where we found an interesting pattern of clustering in their pedicles, and importantly, a closely grouped set of contacts to a nearby SCBC. We have added an example of such a scenario to Figure 4 (Figure 4F) as a proof-of-principle for how such true S-cone clustering may facilitate wiring with SCBCs.

PresentationThe text in general could be expanded to help guide a reader through the figures. I found myself reading portions of the text and figure legends multiple times before I could understand the content of a given figure, and I am not sure I would have had that patience had I not been reviewing the paper. The results are clear and quite beautiful, so guiding someone through them clearly is important! As a general guide I think the paper could expand in length by a factor of ~2.Here are some examples of places I found myself spending a lot of time to understand. (1) Figure 1C. Visually I expected the outer ring to follow from the inner pie chart, and it took a while to realize what was plotted there. (2) Figure 2C. More description in the text of what is plotted here would help. Maybe pick a quadrant and describe each line carefully. (3) A more complete description of how different cone terminals were identified for Figure 3 would be helpful. This is not a complete list – so I would ask the authors to go through the paper and make sure they lead the reader through each figure carefully and not just give the result.

In our original submission, we regrettably omitted detailed explanations in the main text in several locations in the interests of shortening the manuscript to observe the length guidelines for the *eLife* “Short Report” format.

Following the reviewers’ feedback, we have carefully edited the manuscript adding clarification text and new relevant references to better guide the readers through the manuscript and figures. We have also eliminated the outer ring in Figure 1C which, as pointed out, was confusing and did not convey critical information. We added explanations to the main text describing each graph in Figure 2C, and we have expanded upon the method for identifying true S-cone terminals as suggested.

QuantificationIn general, the results are nicely described and then quantified. Figure 3 is an exception, where there are several statements made without any quantitative analysis. It is important, for example, to see some numbers about the types of cone contacts made by the S cone bipolar cells. Similarly, more complete numbers for both convergence and divergence in dorsal and ventral retina are needed (fleshing out details in subsection “2.3. S-cone bipolar cells exhibit a dorsal-ventral gradient with a higher density in the ventral retina.”).

We have now performed quantification for the numbers of SCBCs contacting each true S-cone terminal as well as the numbers of true S-cones contacted by each SCBC. This data has been added Figure 3 and described accordingly in the main text.

Reviewer #2:Nadal-Nicolas et al., present an elegant and important survey of mouse cone distributions across the retina. They show that many S-opsin expressing cones in the ventral retina are in fact "true" S-cones, unlike previously thought. This has important consequences for retinal wiring, colour vision and our understanding of ecological adaptation in vision.I had commented on an earlier version of this work elsewhere. At the time I thought it was excellent and certainly worth publishing in a broad journal, and I continue to think so also now. From that previous version, the authors suitably addressed most of my points raised at the time, accordingly all my remaining points on this new and improved version are minor.

We thank reviewer 2 for the fair and constructive review. We found much of the feedback from the previous reviews helpful and felt the manuscript would be improved if we followed those suggestions.

Reviewer #3:In this study, Nadal-Nicolas and colleagues have used immunohistochemistry and careful anatomical analysis to identify an interesting distribution of cone photoreceptors, that only express short wavelength-sensitive opsin pigment (true S-cones), in mouse retina. Most cones in mouse retina express both the medium wavelength (M) and short-wavelength (S) sensitive opsin. It is well established that there is spatial gradient of opsin expression across the mouse retina – with middle wavelength-sensitive and S opsin enriched in the dorsal and ventral retina respectively. However, previous studies have proposed a uniform distribution of true S-cones across the dorso-ventral axis. This issue has been revisited by Nadal-Nicolas and colleagues where they systematically map the distribution of all cone types i.e. M^+^S^+^, M^+^S^-^ and S^+^M^-^, and surprisingly find that the concentration of true S-cones (S^+^M^-^) is remarkably higher in the ventral retina than in the dorsal retina. When comparing this cone distribution with albino mouse, they find that even though there is robust S opsin expression in the dorsal retina (unlike pigmented mouse), the pattern of true S cone distribution remained mostly unchanged. The authors further tested if the bipolar cells that receive synaptic inputs from the S cones follow the uneven distribution of the true S-cones. They convincingly demonstrate using a novel transgenic mouse, which selectively labels the S-cone bipolar cells, that these bipolar cells mimic the dorsal ventral concentration gradient as the true-S cones. These findings are novel and important for a mechanistic understanding of color vision. The data quality is high and the analysis is robust. Overall, it is well-laid out study with beautiful figures and exciting results.I have a few comments that could potentially add to their findings:i) Is there divergent contacts from a single true-S cone to SCBCs in the ventral retina like in the dorsal retina? Also, it would be informative to know if the SCBCs also get input from the M^+^S^+^ cones compared to S^+^M^-^ cones. How does this change going between dorsal, medial and ventral retina? This is important for making the argument about sensitivity of color opponent signals being higher in the ventral retina.

These are interesting questions. Although convergent as well divergent connections were found between true S-cones and SCBCs in both dorsal and ventral retina, we noted different connectivity patterns. We now quantified the connections between true S-cones and SCBCs in both dorsal and ventral retina. These results agree well with the true S-cone to SCBC ratios calculated from cell densities. Accordingly, both data sets support the presence of a prevalent divergence of true S-cone to SCBC connections in the dorsal retina, in comparison to a prominent convergence of contacts from true S-cones to SCBCs in VN retina. From the data we collected so far, we have not seen evidence that SCBCs contact M^+^S^+^ cones. However, we have observed some “blind” endings that did not reach any cone pedicle as previously reported (Haverkamp et al., 2005; Herr et al., 2003; Kouyama and Marshak, 1992).

ii) The authors have used a robust automated and a manual analysis method to identify and count each of the cone types. In subsection “10.3.6. Image processing: manual and automated whole quantification”, the authors mention using a threshold to create a binary mask and subsequent filtering. It would be great to mention in the methods where the threshold is placed relative to the background noise in terms of standard deviations above noise. Also, could the placement of the threshold be a reason for mis-classifying a small fraction of M^+^S^+^ cones as true S cones since M opsin expression is low?

We were aware of the possible lower M-opsin expression in some mixed cones especially in the ventral retina. To avoid false classification, in addition to automated analysis, we manually identified the true S-cones in both mouse strains. Of note on this point, the importance of removing thoroughly the RPE to obtain optimal opsin immunodetection was a crucial step to enable high fidelity detection of opsins. During manual classification, we only considered true-S cones as those cells completely lacking any detectable M-opsin signal. The mean values of background noise for our data sets were 9.6±1.2% and 15.2±3.2% for S- and M-opsin respectively, and the threshold was applied at 15.7%. Later, the remaining noise signals were excluded consistently by applying the ‘despeckle’ filter and by fixing shape and size parameters.

iii) It is indeed interesting to see the clustered arrangement of the true-S cones in the ventral retina. Is this also a feature of the true M cones (M^+^S^-^) in the dorsal retina? Moreover, is it possible to compare a M cone contacting bipolar cell (any type) and if it has preference of contacting the true M cones vs the M^+^S^+^ cones and if this changes between dorsal and ventral retina. The authors have a great opportunity to also expand on the true M cone distribution and its connectivity across mouse retina.

These are interesting notions. It is possible that M^+^S^-^ cones are sampled by a specific bipolar type, for example, Type 1 bipolar cells have been reported to avoid S-cones (Behrens et al., 2016). However, the vast difference between two strains would implicate very different connectivity patterns for such bipolar cells. We have not examined the connection of M^+^S^-^ cones with other bipolar types, which may involve using a different bipolar mouse line, but it would be interesting to probe in the future, perhaps in albino mice.

iv) The authors have generated a SCBC specific mouse transgenic line which could be very useful for targeting this bipolar cell type for single cell electrophysiology to substantiate the anatomical findings in this study. In particular, the divergent vs convergent inputs from true S cones to SCBCs in dorsal vs ventral retina should have a direct bearing on the strength of the synaptic inputs on SCBCs and this could be validated by measuring short-wavelength light evoked excitatory currents from SCBCs.

We thank for the encouraging suggestions. The SCBC reporter mouse provides a very useful tool for future color vision study in mouse. We are definitely committed to further investigations of the physiology of SCBCs. However, such experiments are probably beyond the scope of this study and will certainly evolve into a separate story.